# Effect of Citrus Pellet on Extrusion Parameters, Kibble Macrostructure, Starch Cooking and In Vitro Digestibility of Dog Foods

**DOI:** 10.3390/ani13040745

**Published:** 2023-02-19

**Authors:** Salvatore Cucinotta, Marianna Oteri, Mayara Aline Baller, Lucas Bassi Scarpim, Camila Goloni, Biagina Chiofalo, Aulus Cavalieri Carciofi

**Affiliations:** 1Department of Veterinary Sciences, University of Messina, 98168 Messina, Italy; 2Faculdade de Ciências Agrárias e Veterinárias, Universidade Estadual Paulista (UNESP), Jaboticabal 14884-900, SP, Brazil

**Keywords:** citrus pulp pellet, extrusion, fiber, mechanical energy, processing efficiency

## Abstract

**Simple Summary:**

Recently, there has been widespread social and environmental pressure for the efficient reuse of agricultural industry residues due to the global intensification of food production, which creates large quantities of food co-products. Citrus Pulp Pellet (CPP) is the solid waste part of orange juice production, characterized by a good proportion of soluble and fermentable fiber. The aim of the study was to evaluate the influence of increasing amounts of CPP on the extrusion process and kibble characteristics as well as on the digestibility of dog foods. Five diets with different CPP inclusions (0%, 5%, 10%, 15% and 20%) were developed for adult dogs and produced in a single screw extruder. The inclusion of CPP in the formula had an impact on the extrusion traits, influencing the processing parameters and the characteristics of the final product; therefore, the beneficial effects of this ingredient cannot be generalized. However, recycling and proper use of co-products in pet foods improve sustainable agriculture by transforming low-quality co-products into high-quality foods. This, in compliance with current legislation, we strongly encourage the food industry to find new end-uses for refusals such as the CPP exemplified in the present study.

**Abstract:**

Fiber supplemented extruded foods are produced by pet food companies to help with several specific health conditions. The fiber material, however, is difficult to incorporate efficiently into dry kibble diets for dogs. The aim of the study was to evaluate the influence of citrus pulp pellet (CPP), the solid waste part of the production of orange juice characterized by a good proportion of soluble and fermentable fiber, on extrusion traits, kibble macrostructure, starch gelatinization and in vitro digestibility of dog foods. A control formula (CO) was developed for dogs. CPP was added to CO at different inclusion levels: 5%, 10%, 15% and 20%. Foods were extruded in a single screw extruder using two different die diameters (d_d_): 5 mm and 7 mm. CPP inclusion with 5 mm d_d_ did not affect bulk and piece density and resulted in a lower impact on kibble expansion; It also resulted in greater starch gelatinization and kibble expansion compared to the 7 mm d_d_ configuration (*p* < 0.01). In addition to the nutritional implications, recycling and proper uses of this material exemplified in the present study by the exploration of CPP as a fiber source to dogs, this method can improve sustainable agriculture by transforming low-quality materials into high-quality foods.

## 1. Introduction

The production of citrus fruits has been widespread in the Mediterranean area since ancient times [1]. Citrus fruits are processed to obtain oil and juice, with the consequent production of huge quantities of by-products of industrial process, including peel (60–65%), pulp (30–35%) and seeds (0–10%) [2]. These represent a potential important resource due to their nutritional value and bioactive compounds [1]. Citrus pulp pellets (CPP) consist of the solid waste parts obtained after fruit pressing, composed of orange seeds, flavedo, albedo and central core; they also include the solids retained in the juice filtration process, which are rich in fermentable carbohydrates mostly composed of soluble dietary fiber, such as pectin. CPPs represent an interesting alternative fiber source in animal feed [3], including dog food [4,5]. The citrus pulp pellet, rich in soluble and fermentable complex carbohydrates, represents an alternative ingredient for promoting growth of the microbiota by producing, in the distal part of the gastrointestinal tract, a high quantity of short-chain fatty acids, in particular acetate, propionate and butyrate [4]. This indicates a potential prebiotic role of these substrates [6]; in particular, the elevated production of butyrate could be useful for the colonic epithelium as the main energy source for cell growth and differentiation and enterocytes homeostasis [4,7]. Another specialized co-product of the orange juice industry is the orange fiber, a semi-purified material selectively composed of the material obtained from the juice filtration process. The material has a very high content of soluble and fermentable fiber and promotes a good fermentation profile when added to dog [8] and cat [9] foods.

This fiber material, however, is difficult to incorporate efficiently into dry kibble diets for dogs [10]. The extrusion process requires a combination of moisture, shear, temperature and pressure, which are applied to the mass in a continuous and short-time process; the raw material mixture is forced into a die with specially designed openings [11]. The extrusion process promotes starch gelatinization, denaturation of protein, lipid modification, enzyme inactivation and reduction of microbial viability of the raw materials [12,13]. In particular, the starch content is plasticized in the extruder barrel. Upon exiting the barrel at atmosphere pressure, it expands, creating a cellular structure which affects the kibble shape and texture [14].

The characteristics of the CPP included in the formula, in particular its fiber content, have an impact on the extrusion traits, influencing the processing parameters and potentially altering the characteristics of the final product [4,10]. Fiber materials may reduce the mechanical energy transference, which is composed of mechanical and thermal energy as measured by the specific mechanical energy (SME) and specific thermal energy (STE) input, respectively [10]. Fiber materials may also reduce the viscosity development and cell structure formation, which may result in harder and less expanded kibbles; therefore, the beneficial effects of this ingredient cannot be generalized [15].

Therefore, the hypothesis of this study was that CPP could be a viable alternative to traditional fibrous ingredients, but the inextensible and fibrous characteristics of this material will affect energy transference during the extrusion process, reducing starch cook, kibble expansion and structure formation: this can ultimately affect the diet palatability and utilization. Thus, the objective of the present study was to determine the effects of increasing amounts of CPP and two extruder open areas on the extrusion parameters, kibble macrostructure, starch gelatinization and in vitro digestibility of dog foods.

## 2. Materials and Methods

### 2.1. Fiber Ingredient and Diet Formulation

The CPP used in this study was donated by Citrosuco Paulista S.A. (Araras, Sao Paulo, Brazil). In the orange juice extraction process, a solid material composed of orange seeds, albedo, flavedo and the fruit central core is retained during filtration, representing up to 50% of the fruit weight. This material is mixed with hydrated lime and pelletized to obtain the CPP. Table 1 shows the chemical composition of the CPP used in the study.

To evaluate CPP inclusion, a control formula (CO) with maize and poultry by-product meal as the main ingredients and without added fiber source was formulated (Table 2). To produce the experimental diets, CPP was added to CO, replacing maize in the following percentages: 5% (CPP5), 10% (CPP10), 15% (CPP15) and 20% (CPP20). Diets were balanced to adult dogs and followed the European Pet Food Industry Federation Nutritional Guidelines [16]. 

### 2.2. Diet Preparation

To prepare the diets, the ingredients were weighed and mixed in a paddle mixer. Subsequently, the mixture was ground using a hammermill equipped with a 1 mm screen sieve (Sistema Tigre de Mistura e Moagem, Tigre, Sao Paulo, Brazil). A single-screw extruder with a screw diameter of 80 mm and a processing capacity of 250 kg/h (MEX 250, Manzoni, Campinas, Brazil) was used to produce the diets. The extruder was equipped with two extruder die diameters (d_d_): a d_d_ of 5 mm (resulting in an open die area of 19.6 mm^2^) and a d_d_ of 7 mm (resulting in an open die area of 38.5 mm^2^).

The extruder open area (mm^2^/ton/h) was calculated as:Extruder open area = die open area (mm^2^)/extruder output mass (ton/h)

The extruder screw had five sections: initial—single flight and no steam lock; second—single flight and small steam lock; third—double flight uncut and small steam lock; fourth—double flight uncut and medium steam lock; fifth—double flight cut cone. For all treatments, the extruder screw speed was set to 643 rpm. No water or steam was injected in the extruder barrel. 

The extruder was equipped with a double-shaft preconditioner with differential diameter. The small cylinder was set at 60 rpm and the large cylinders at 30 rpm, obtaining an average mass residence time of 182.7 s. Thermal energy was applied directly by infusing steam into the mass, resulting in a mass temperature outside the preconditioner of 95.4 °C ± 1.26. A volumetric delivery system operating at an approximate feed rate of 170 kg/h delivered food into the system. Water was injected at a rate of about 40 kg/h into the preconditioner.

The product moisture at the preconditioner had an average value of 25.9% (±1.44). The product flow rate at the preconditioner was 220.5 kg/h (±25.6) for d_d_ 7 mm diets and 198.5 kg/h (±6.51) for d_d_ 5 mm diets. During the production of the treatments, some fluctuations on the applications of specific thermal energy (STE) were detected, with an average value of 80.28 kW-h/ton (±17.5) for d_d_ 7 mm diets and 87.03 kW-h/ton (±21.4) for d_d_ 5 mm diets. All treatments (CO, CPP5, CPP10, CPP15 and CPP20) were processed separately at two different days for each d_d_, for a total of two extrusion days for each combination of CPP level and d_d_. The CO diet was fed to the extruder at the beginning of both days to warm up the equipment and reach steady state before starting treatment production. At least 30 min were waited after the stabilization of the equipment before the collection of the experimental data was started. No further changes were made to the independent process parameters (feeder screw speed, extruder screw speed, water and steam infusion and cutting knife speed) after this point. Thus, all the variations recorded on the variable extrusion parameters and on the kibble formation could be attributed to the addition of CPP and the d_d_ extruder. 

The following parameters were recorded at each 15 min interval, with at least four measurements per diet and per d_d_, for a total of eight observations per treatment (4 repetitions × 2 processing days): mass temperature at preconditioner exit, engine amperage, mass temperature before extruder die, mass temperature at extruder exit, mass pressure before extruder die and extruder output mass. Samples of the mass were also collected at the preconditioner and extruder exit, and they were dried and stored at −20 °C until further analysis. The bulk density of the kibbles was measured after the extruder (the weight corresponds to the volume of 1 L of kibbles). Other parameters were also recorded at 15 min intervals: room temperature, working water temperature, mash feed temperature and steam pressure. To remove moisture, the kibbles were dried after the extruder in a forced air dryer at 105 °C for 20 min. They were then coated with a liquid palatant and poultry fat according to the formulation.

The mean geometric diameters of each raw material mixture after grinding were established according to Zanotto and Bellaver [17] and calculated using the Gransuave program (Embrapa, Brasilia, Brazil). The particle sizes of the diets for d_d_ 7 mm and d_d_ 5 mm, respectively, were CO: 285 and 298 µm; CPP5: 278 and 263 µm; CPP10: 276 and 309 µm; CPP15: 278 and 299 µm; and CPP20: 281 and 298 µm. There was an overall mean geometric diameter of 287 µm.

### 2.3. Chemical Analyses

Before analyses, the kibbles were ground (Mod MA-350, Marconi, Piracicaba, Brazil) in a cutting mill with a 1 mm screen sieve. The following procedures described by the Association of Official Analytical Chemists [18] were adopted: method 934.01 for dry matter (DM); method 942.05 for ash content; method 990.03 for crude protein using a LECO nitrogen/protein analyzer (FP-528, LECO Corporation, Saint Joseph, MI, USA); method 954.02 for acid hydrolyzed ether extract; and method 978.10 for crude fiber (CF). Hendrix’s method [19] was adopted to analyze the starch content. The starch content was measured using the amyloglucosidase method [20] with modifications proposed by Sá et al. [21]. A bomb calorimeter (IKA C2000 Basic, IKA-Werke GmbH & Co. KG, Staufen, Germany) was used to determine the gross energy (GE). The phosphorus content was determined by a spectrophotometer (model B442; Micronal, Brazil) using the vanadate-molybdate method and calcium by atomic absorption (model GBC-932 AA; Scientific Equipment PTY Ltd., Melbourne, Australia) [18]. The in vitro digestibility of organic matter was determined following the method proposed and validated for dogs by Hervera et al. [22]. The method simulates the digestion process in two steps: stomach and small intestine. Briefly, samples are first incubated for 2 h with phosphate buffer (0.1 M, pH 6), hydrochloric acid (0.6 M, pH 1) and pepsin (10 mg in 1 mL of solution; Sigma-Aldrich Brasil, Cotia, Brazil) at pH 2, adjusted with 1 M hydrochloric acid and 1 M sodium bicarbonate solutions. A second incubation is performed with pancreatin solution (100 mg in 1 mL of solution; Sigma-Aldrich Brasil, Cotia, Brazil) for four hours, adjusting the pH to 6.8 pH with phosphate buffer (0.2 M, pH 6.8) and 0.6 M sodium hydroxide. After the two steps of the in vitro digestion, the samples were filtrated, washed with acetone, dried, and subjected to ash quantification in a muffle furnace. The in vitro digestibility of organic matter (OM) was then calculated as:In vitro digestibility OM = ((incubated OM − residual OM)/incubated OM) × 100

### 2.4. Specific Mechanical Energy and Specific Thermal Energy Calculations

The specific mechanical energy (SME; kW-h/ton) applied for each experimental treatment was calculated using the equation proposed by Riaz and Aldrich [23]:SME = (√3 × Voltage × (At − Av) × cosFi)/M
where: 

Voltage = 220 V; 

At = torque load working amperage (A); 

Av = no torque load working amperage (A); 

cosFi = power factor; 

M = mass flow rate from extruder (kg/h).

The specific thermal energy (STE; kW-h/ton) was calculated as the net thermal energy introduced through the absorption of steam and water from the mass inside the preconditioner, divided by the raw material throughput (ton/h) [11]. The net steam absorption (kg/h) was calculated from the mass balance according to Riaz [11]. The corresponding thermal energy was calculated by multiplying the steam enthalpy (kJ/kg) from steam tables and adjusting for the average mass temperature inside the preconditioner, as described in detail by Pacheco et al. [24].

The Total Specific Energy (TSE; kW-h/ton) was obtained by the sum of the SME and STE.

### 2.5. Kibble Traits and Macrostructure

The radial expansion ratio (RE), piece density (ρ; kg/m3) and specific length (lsp; mm/g) were determined in 20 representative kibbles of each experimental treatment. To obtain these parameters, the length (le), the diameter (de) and the mass (me) were measured and the following equation was used:RE = de^2^/d_d_^2^
l_sp_ = le/me
ρ = 4 me/(π × de^2^ × le)
where: d_d_ = die diameter.

The kibble was stabilized in an oven at 35 °C for 24 h to exhibit similar moisture content (ETS Model 532, Systems Electro-Tech, Inc., Glenside, PA, EUA), and the extrudate cutting force was determined using a texture analyzer (Texture Analyzer TAX/T2I–Stable Micro Systems Ltda, Godalming, UK). Twenty kibbles were weighed and analyzed with a load cell of 50 kgf, a penetration distance of 10 mm and a speed of 2.0 mm/s, using a Warner Bratzler Knife [10]. To avoid deviations, only kibbles whose diameter corresponded to the mean diameter of the treatment were used.

As a complementary evaluation, electron microscope images of the kibbles were taken at the Chemistry Institute of UNESP, Araraquara, Brazil. A scanning field emission electron microscope (JEOL, JSM-7500F; Miaka, Tokyo, Japan), adjusted to 20 kV, was used. The examiner of the images was blind to the experimental groups. Only qualitative analysis was performed and without statistical evaluation. To evaluate the internal structure, the kibbles were cut along the medial axis.

### 2.6. Statistical Analysis

Data of the extrusion test were analyzed as a factorial arrangement of 5 (formulas) × 2 (die diameters) for a total of 10 treatments. The study followed a randomized design, with two days of extrusion (plot) and four repetitions per day of extrusion (each 15-min sampling interval) in a split plot over time design. The experimental unit was considered to be the day of extrusion and the sample and processing data collected at each 15-min interval subplots. For kibble macrostructure and cutting force, the experimental unit was one kibble, with 20 repetitions per treatment. Data were submitted to analysis of variance, and model sums of squares were separated into the effect of diet (CO, CPP5, CPP10, CPP15, CPP20), d_d_ (d_d_ 7 mm and d_d_ 5 mm) and their interaction (diet × d_d_). When differences were found in the F test, the means were compared by orthogonal polynomial contrast considering the CPP inclusion level. All data were found to comply with ANOVA model (error normality was evaluated by the Cramer-von Mises test and homoscedasticity of variance by the Levene test) and analyzed using GLM procedure of SAS software (vers. 9.1) [25]. The *p* significance level was set at 0.05.

## 3. Results

The CPP had low starch, moderate protein and high total dietary fiber content, which were expected, given the ingredients characteristics (Table 1). During the formulation of the treatments, the only change was the substitution of maize by increasing additions of CPP, without altering the other raw materials’ inclusion. This could explain the differences in the chemical composition among treatments, as starch decreased and crude fiber increased with the increasing CPP inclusion (Table 3). The content of the other nutrients was similar between treatments.

An effort was made during the extrusions runs to not change software parameters such as in-barrel moisture and product flow rate (Table 4). However, while changes on in-barrel moisture (*p* < 0.05) were observed, they were small and had little potential to influence treatment results. Product flow rates differed between different values of d_d_ (*p* < 0.01). The product flow rate showed no differences at the 5 mm d_d_ for different CPP inclusions (*p* > 0.05). However, it increased quadratically while producing diets with the 7 mm d_d_. This may have impacted the observed parameters and should be considered in the data interpretation. As proposed on the experimental design, the differences in d_d_ caused the extruder open area to be more than 50% greater when the 7 mm d_d_ was used in comparison to the 5 mm d_d_ (*p* < 0.01). The higher restriction to the extruder mass flow explains the greater application of specific mechanical energy observed for the 5 mm d_d_ compared to the 7 mm d_d_ extruder configuration (*p* < 0.01). 

The amount of CPP inclusion affected the mass pressure before the die, which increased for 7 mm d_d_ and reduced linearly for the 5 mm d_d_ (*p* < 0.01) (Table 4). The CPP inclusion induced a quadratic decrease on motor amperage (*p* < 0.01), a quadratic (7 mm d_d_) or linear (5 mm d_d_) increase on mass temperature before the die (*p* < 0.01), and a quadratic decrease on SME application to the mass for both die diameters (*p* < 0.01) (Table 4).

All of these alterations affected kibble characteristics and cooking level, as can be seen in Table 5. When recipes were processed with the 7 mm d_d_, piece volume, radial expansion and specific length were all decreased quadratically (*p* < 0.01), while kibble bulk density and piece density increased quadratically with increasing CPP inclusion (*p* < 0.01). 

With the greatest restriction on mass flow and the greatest application of mechanical energy with the 5 mm d_d_ configuration, although the linear reduction in piece volume, radial expansion, and specific length (*p* < 0.05) decreased, kibble bulk density and piece density did not change (*p* > 0.05). 

The impact of CPP addition on kibble parameters was significantly (*p* < 0.01) lower, as observed for the interaction between d_d_ and CPP addition. For example, for the 5 mm d_d_, only a minor (not significant) numerical increase of 11 g/L on kibble bulk density after CPP addition was observed, but this increase was of 56 g/L in the 7 mm d_d_ processing (*p* < 0.01). The piece volume decreased by 42% when comparing CO and CPP20 on the 7 mm d_d_, but only decreased by 15% on the 5 mm d_d_ configuration (*p* < 0.01). Another example is the cutting force, or kibble hardness, that increased quadratically in both d_d_ (*p* < 0.01). However, kibbles produced with the 7 mm d_d_ configuration were harder (*p* < 0.01).

For the starch gelatinization (Table 5), higher values were observed for treatments processed with 5 mm compared to 7 mm d_d_ (*p* < 0.01); this was also a consequence of the higher SME applied. However, starch gelatinization decreased quadratically for the 7 mm d_d_ and linearly for the 5 mm d_d_ extruder configurations with the increase of CPP (*p* < 0.01). Finally, the in vitro digestibility of OM (Table 5) was also influenced by both CPP inclusion and extruder die (d_d_). It was higher for 5 mm d_d_ than for 7 mm d_d_ (*p* < 0.05), and decreased quadratically for the 7 mm d_d_ and linearly for the 5 mm d_d_ after CPP inclusion (*p* < 0.01). 

The internal scanning electron micrographs of the kibbles are shown in Figure 1 and Figure 2. The qualitative evaluation of the images showed larger cells with larger internal areas in the CO diet (open space of the internal areas is larger) for both d_d_. With the CPP inclusion, the cells became progressively smaller, with a small internal area and thicker walls. It is also possible to verify that the number of cells per kibble increased with the addition of CPP. This, however, was subjectively more pronounced in the 7 mm d_d_ open area when practically no cellular structure was observed after 10% of CPP inclusion (Figure 1). The impact of CPP was apparently lower in the 5 mm d_d_, as cellular structure is observed for all treatment formulations (Figure 2).

## 4. Discussion

The observed impact of CPP addition on kibble formation and extrusion parameters has been previously demonstrated after the inclusion of fiber sources in human [26,27,28] and dogs’ extruded snacks [4]. The observed reduced resistance to mass flow can be attributed to the plastic and elastic nature of CPP fiber, composed largely of pectin and other soluble compounds [29], which may deform to pass through the extruder die. The parallel reduction in the starch content and any other components in the mass that are thermoplastic in nature and induce greater resistance to mass flow after being gelatinized could also explain this reduction in motor amperage. It is also possible that the reduction in starch content increased the water available during extrusion, thus increasing the dough fluidity [30]. The reduction of mechanical energy consumption with the CPP inclusion has positive implications in the production cost and indicates a lower utilization of electric energy. Furthermore, the lower input of mechanical energy may be associated with less wear stress on the equipment, reducing the frequency of part replacement [31,32]. Although this same result was previously observed by Pacheco et al. [4] for the addition of CPP to dog foods, it cannot be generalized; Monti et al. [10] found an increase in specific mechanical energy after adding guava fiber, sugarcane fiber and wheat bran to pet food. One possible explanation for these different results is the physicochemical nature of these three fiber sources, which were basically insoluble, filamentous (sugarcane fiber and wheat bran) and with a high cellulose content structuring the cell wall of the plant. A possible more rigid and non-deformable characteristic of these fibers can increase the friction and resistance to mass flow through the extruder die. These differences suggest that fibrous material characteristics are important for their effect on mass flow resistance and extrusion traits, reinforcing the need for studies about the role of fiber in pet food processing. 

The kibble macrostructure and starch cook, on the other hand, were negatively influenced in treatments with fiber supplementation, as previously observed in dog foods [4,10]. The starch fraction of the recipe functions as a thermoplastic polymer during extrusion; when there is sufficient water, energy and time during processing, the starch granules lose their crystallinity, swells and disrupts, forming an amorphous mass that binds all the food components in a continuous structure [33]. In the present study, when a higher restriction was applied in the extruder die (5 mm d_d_), the increase in mass flow resistance increased energy application, restoring starch transformation and cooking.

Lower energy implementation probably contributed to lower starch gelatinization and poor kibble expansion in fiber supplemented foods [30,34]. Additionally, the added fiber, mostly composed of soluble dietary fiber such as pectin, may have retained water [35,36], which would then become less available for starch hydration and gelatinization. Therefore, further studies may be needed to compensate for this by, for example, changing processing conditions (such as water and steam addition or extruder screw speed) or hardware conditions (such as screw or die plate configuration) to promote sufficient energy transference to obtain adequate cooked material and structured kibbles [24].

The shift towards longitudinal rather than radial expansion and the increase in kibble density were already described after the inclusion of the fiber source [4,10]. The addition of fiber material can also induce the formation of kibbles with thicker and smaller cell size [37], as observed in the scanning electron micrographs in the present study. These changes in cell structure could be responsible for the higher cutting force (kibble hardness) with CPP addition. Smaller cells with thicker walls increase the material strength against the rupture of the molten starch mass, highlighting the significant effects of the fiber on kibble structure [38,39,40]. Due to its characteristic structure, the fiber can conduct water vapor out of the kibble, reducing the flash off action on cell formation, which is considered an additional factor for poor cell structure formation and less expansion [14,37].

However, it is noteworthy that reducing the extruder open area by approximately 40%, comparing the extrusions with 5 mm d_d_ or 7 mm d_d_, induced a higher SME transference that was sufficient in increasing kibble expansion and starch cook, substantially reducing the cutting force. Furthermore, kibble bulk density increased after CPP addition when using the 7 mm d_d_, but remained unchanged for the 5 mm d_d_: this highlights that it is possible to compensate for the effects of fiber on kibble structure by altering extrusion processing conditions such as die diameter.

The in vitro digestibility coefficient of OM was higher with higher energy transference, lower extruder d_d_ and higher starch gelatinization. It decreased with increasing CPP inclusion levels. This is explained as the adopted in vitro method uses pepsin to simulate gastric digestion and pancreatin to simulate small intestine digestion. The α-amylase in the pancreatin solution only degrades gelatinized starch, so it was possible to observe it in relation to the amount of starch (inversely related to the level of CPP inclusion) and the cooking extent (positively associated to the SME application). The effect of CPP inclusion is explained as the fiber is not digested by mammalian enzymes.

Several implications for the interference of fiber addition on expansion, cell structure formation and hardness increase should be considered. As subjectively observed in scanning electron micrographs, the internal area of the cell decreased and the cell walls thickened with increasing CPP inclusion. However, this was less evident after increasing die restriction and, subsequently, SME. During coating, the migration of fat from the surface to the internal areas of the kibble may be reduced due to the smaller internal open space of the kibble. Lower porosity and thicker cell walls can also inhibit the migration of fat remaining on the surface after coating. The fat distribution inside the kibble is considered important for the food acceptance and palatability and may also explain the lower acceptance of fiber-supplemented diets [41]. Another problem is that fat that remains on the surface can migrate to the packaging, interfering with the food appearance and nutrient content. Adjustments to the coating system or operating conditions may be necessary to reduce these problems, such as a longer residence time, more efficient mixing or even the use of special vacuum coating systems.

All of these changes in texture and shape can affect the palatability of dog food. However, the influence of physical attributes of kibble is poorly studied in dogs and its effect on acceptability and intake is poorly understood [41]. Intake behavior can also be influenced, as these physical alterations can change the sensation and kinetics of chewing.

These possible effects on palatability, however, must be considered for each specific fiber source. For example, although orange fiber addition reduced the application of specific mechanical energy and expansion and increased hardness, it resulted in more palatable kibbles for dogs than the un-supplemented control diet [4].

All these results are important in a scenario of increasing the use of fiber supplemented diets for dogs and cats [42]. Common conditions as obesity and old age modify glucose tolerance and insulin sensitivity of animals [43,44]. Senescence alterations of microbiota and intestinal function may require nutritional interventions with fiber-supplemented diets [45,46]. The gut microbiome benefits of CPP addition were previously evaluated by Pacheco et al. [4], who reported a high fermentability of CPP dietary fiber by the colon microbiota of dogs with production of a high amount of short-chain fatty acids, indicating a potential prebiotic role of CPP soluble fiber. Thus, the implications of fiber on processing efficiency, kibble formation and starch cook described here may allow the industry to establish more efficient processing parameters to develop fiber supplemented diets.

Thus, the implications of fiber on processing efficiency, kibble formation and starch cook described here may allow the industry to establish more efficient processing parameters.

In addition to the nutritional and processing implications of using fiber sources, industrial ecology and the circular economy are considered the leading principles for eco-innovation focused on a ‘zero waste’ society. Recently, there has been widespread social and environmental pressure for the efficient reutilization of agricultural industry residues [47] due to the global intensification of food production, which creates large quantities of food co-products [48]. This may represent a loss of valuable biomass and nutrients [3,48,49,50]. Recycling and proper utilizing co-products in pet foods, such as CPP exemplified in the present study, also improves agriculture profitability by transforming low-quality materials into high-quality foods [51,52]. Co-products are not to be considered waste, but rather, raw materials obtained from agriculture and/or industrial processes [48]; this complies with current legislation, which strongly encourages the food industry to find new end-uses for refusals [53].

## 5. Conclusions

The CPP included in the dog food formulation reduced mechanical energy transference and kibble expansion. It altered cell structure formation with the development of smaller cell sizes with thicker walls. This reduction in processing efficiency also reduced the starch cook. The reduction in mechanical energy transference could be attributed to the soluble nature of CPP fiber, mainly consisting of pectin, which facilitates mass flow through extrusion. This effect was partially overcome with the reduction of the extruder open area, which allowed for greater transference of mechanical energy, reduced the impact of the fiber on expansion and led to more starch cook.

## Figures and Tables

**Figure 1 animals-13-00745-f001:**
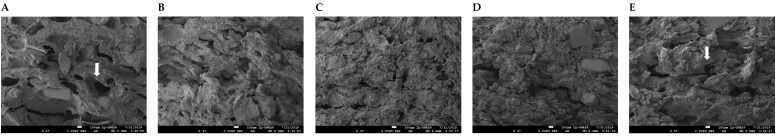
Scanning electron micrograph of kibbles produced with different inclusions of citrus pulp and with a die diameter (d_d_) of 7 mm. (**A**–**E**) correspond to the internal area of the CO, CPP5, CPP10, CPP15 and CPP20 diets, respectively (Increase of 37×). CO = control food, without added citrus pulp pellet; CPP5 = addition of 5% citrus pulp pellet; CPP10 = addition of 10% citrus pulp pellet; CPP15 = addition of 15% citrus pulp pellet; CPP20 = addition of 20% citrus pulp pellet. Arrows indicate internal cells: in **A**, the cells are formed with thick cell walls; in **E**, a continuous mass without identifiable cell structure is observed.

**Figure 2 animals-13-00745-f002:**
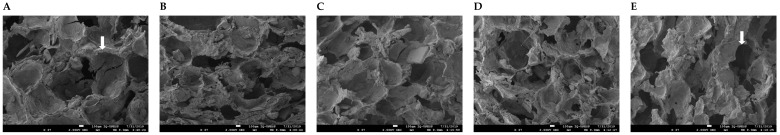
Scanning electron micrograph of kibbles produced with different inclusions of citrus pulp and with a die diameter (d_d_) of 5 mm. (**A**–**E**) correspond to the internal area of the CO, CPP5, CPP10, CPP15 and CPP20 diets, respectively (Increase of 37×). CO = control food, without added citrus pulp pellet; CPP5 = addition of 5% citrus pulp pellet; CPP10 = addition of 10% citrus pulp pellet; CPP15 = addition of 15% citrus pulp pellet; CPP20 = addition of 20% citrus pulp pellet. Arrows indicate internal cells: in **A**, the cells are well formed, large and with thin cell walls; in **E**, smaller cells with thick cell walls are observed.

**Table 1 animals-13-00745-t001:** Chemical composition of the citrus pulp pellet used in the study.

Citrus Pulp Pellet ^1^	g/kg, as Fed ^2^
Moisture	107.2
Crude protein	272.0
Starch	33.9
Crude fat	56.0
Crude fiber	157.2
Total dietary fiber	502.2
Ash	94.0
Calcium	22.5
Phosphorus	0.05

^1^ Citrus pulp pellet was provided by Citrosuco Paulista, Araras, São Paulo, Brazil. ^2^ Analyzed in duplicate.

**Table 2 animals-13-00745-t002:** Ingredient composition of the experimental diets (g/kg, as fed basis).

Item	Experimental Formulations ^a^				
	CO	CPP5	CPP10	CPP15	CPP20
Dry formula					
Maize	570.2	520.2	470.2	420.2	370.2
Poultry by-product meal	300	300	300	300	300
Citrus pulp pellet ^b^	-	50	100	150	200
Potassium chloride	5.3	5.3	5.3	5.3	5.3
Sodium chloride	5	5	5	5	5
Vitamin and mineral mix ^c^	5	5	5	5	5
Choline chloride	2.5	2.5	2.5	2.5	2.5
Antioxidant ^d^	0.5	0.5	0.5	0.5	0.5
Mold inhibitor ^e^	1	1	1	1	1
D-L methionine	0.5	0.5	0.5	0.5	0.5
Coating					
Poultry fat	90	90	90	90	90
Palatant enhancer ^f^	20	20	20	20	20

^a^ CO = Control food, without added citrus pulp pellet (formulated to be complete and balanced for adult dog maintenance according to FEDIAF, 2019); CPP5 = addition of 5% citrus pulp pellet; CPP10 = addition of 10% citrus pulp pellet; CPP15 = addition of 15% citrus pulp pellet; CPP20 = addition of 20% citrus pulp pellet. ^b^ Citrosuco Paulista, Araras, Brazil. ^c^ Rovimix, DSM Produtos Nutricionais Brasil S.A., Jaguaré, Brazil. Added per kg of food: Vitamin A, 18,750 IU; Vitamin D3, 1500 IU; Vitamin E, 125 IU; Vitamin K3, 1.5 mg; Vitamin B1, 5 mg; Vitamin B2, 16.25 mg; Pantothenic Acid, 37.5 mg; Vitamin B6, 7.5 mg; Vitamin B12, 45 mcg; Vitamin C, 0.125 g; Nicotinic Acid, 0.0625; Folic Acid, 0.75 mg; Biotin, 0.315 mg; Choline, 0.625 g; Iron, 0.1 g; Copper, 9.25 mg; Manganese, 6.25 mg; Zinc, 0.15 g; Iodine, 1.875 mg; Selenium, 0.135 mg. ^d^ Mold-Zap Citrus, Alltech do Brasil Agroindustrial Ltda., Araucária, Brazil. ^e^ Banox, Alltech do Brasil Agroindustrial Ltda., Araucária, Brazil. ^f^ D’TECH 10L, Palatabilizante Líquido, SPF do Brasil, Descalvado, Brazil.

**Table 3 animals-13-00745-t003:** Analyzed chemical composition (g/kg, DM-basis) and gross energy (kcal/g) of dog foods with different inclusions of citrus pulp pellet processed with two extruder die diameters.

Item		Experimental Formulations ^a^				
		CO	CPP5	CPP10	CPP15	CPP20
	d_d_ ^b^					
Dry Matter						
	7 mm	938	946	954	948	952
	5 mm	949	950	947	951	947
Starch						
	7 mm	342	320	315	294	267
	5 mm	350	346	296	285	252
Crude Protein						
	7 mm	286	281	274	263	271
	5 mm	295	272	274	281	275
Fat						
	7 mm	110	116	115	112	96
	5 mm	105	112	122	119	100
Crude Fiber						
	7 mm	26.8	24.6	31.1	40.2	52.1
	5 mm	26.0	24.1	30.9	40.2	54.5
Ash						
	7 mm	97.2	80.8	85.8	87.6	95.8
	5 mm	81.1	104.7	87.1	91.6	93
Gross Energy						
	7 mm	4211	4309	4253	4200	4139
	5 mm	4234	4405	4298	4226	4307

^a^ CO = control food, without added citrus pulp pellet; CPP5 = addition of 5% citrus pulp pellet; CPP10 = addition of 10% citrus pulp pellet; CPP15 = addition of 15% citrus pulp pellet; CPP20 = addition of 20% citrus pulp pellet. ^b^ die diameters = d_d_ of 5 mm (resulting in a die open area of 19.6 mm^2^) and d_d_ of 7 mm (resulting in a die open area of 39.5 mm^2^).

**Table 4 animals-13-00745-t004:** Processing parameters of dog foods with different inclusions of citrus pulp pellet processed with two extruder die diameters.

Item	d_d_ ^e^	Experimental Formulations ^a^					SEM ^b^	*p* Value ^c^			Contrast ^d^	
		CO	CPP5	CPP10	CPP15	CPP20		d_d_	CPP	d_d_ × CPP	Linear	Quadratic
In barrel moisture (%)												
	7 mm	24.6	23.4	23.1	25.1	25.5	0.16	<0.001	<0.001	<0.001	<0.001	<0.001
	5 mm	27.8	24.2	28.7	27.6	27.2	0.17				0.141	<0.001
Product Flow Rate (kg/h)												
	7 mm	204.3	227	218.5	221.8	225.8	1.16	<0.001	<0.001	<0.001	<0.001	<0.001
	5 mm	174.3	161	174.2	171.0	173.8	1.2				0.053	0.056
Extruder open area (mm^2^/ton/h)												
	7 mm	221.7	169.6	174.7	173.6	170.5	0.66	<0.001	<0.001	<0.001	<0.001	<0.001
	5 mm	112.7	122.2	112.7	114.9	113.1	0.68				0.047	0.005
Specific mechanical energy application (kW-h/ton)												
	7 mm	12.5	9.77	10.4	8.87	8.35	0.13	<0.001	<0.001	<0.001	<0.001	<0.001
	5 mm	15.7	13.7	12.8	10.7	9.6	0.13				0.009	<0.001
Pressure before extruder die (bar)												
	7 mm	22.4	22.7	25.2	23.6	23.0	0.36	<0.001	<0.001	<0.001	0.262	0.012
	5 mm	18.8	18.9	16.2	18.9	15.5	0.34				0.001	0.428
Motor amperage (A)												
	7 mm	37.8	37.7	37.6	36.8	36.5	0.15	<0.001	<0.001	<0.001	<0.001	0.007
	5 mm	36.4	37.2	37.7	36.3	35.8	0.15				0.006	<0.001
Mass temperature before extruder die (° C)												
	7 mm	135.7	144.3	142.0	140.8	142	0.57	<0.001	<0.001	<0.001	<0.001	<0.001
	5 mm	135.3	124.3	139.3	136.5	137.5	0.57				<0.001	0.128

^a^ CO = control food, without added citrus pulp pellet; CPP5 = addition of 5% citrus pulp pellet; CPP10 = addition of 10% citrus pulp pellet; CPP15 = addition of 15% citrus pulp pellet; CPP20 = addition of 20% citrus pulp pellet. ^b^ SEM = standard error of the mean (n = 8). ^c^ d_d_ = diet diameter; CPP = citrus pulp pellet. ^d^ Linear and quadratic effect of citrus pulp pellet addition. ^e^ die diameters = d_d_ of 5 mm (resulting in a die open area of 19.6 mm^2^) and d_d_ of 7 mm (resulting in a die open area of 38.5 mm^2^).

**Table 5 animals-13-00745-t005:** Kibble macrostructure, starch gelatinization and in vitro digestibility of organic matter of dog foods with different inclusions of citrus pulp pellet processed with two extruder die diameters.

Item	d_d_ ^e^	Experimental Formulations ^a^					SEM ^b^	*p* Value ^c^			Contrast ^d^	
		CO	CPP5	CPP10	CPP15	CPP20		d_d_	CPP	d_d_ × CPP	Linear	Quadratic
Piece volume (mm^3^)												
	7 mm	49.3	39.7	31.1	31.6	34.8	0.53	<0.001	<0.001	<0.001	<0.001	<0.001
	5 mm	43.1	41.1	40.4	38.9	37.4	0.55				0.024	0.945
Radial expansion rate												
	7 mm	10.7	7.56	6.31	6.1	6.16	0.13	<0.001	<0.001	0.003	<0.001	<0.001
	5 mm	13.9	12.7	11.8	10.5	9.8	0.14				<0.001	0.670
Specific length (mm/g)												
	7 mm	2.31	2.16	1.98	2.08	2.18	0.02	<0.001	<0.001	<0.001	<0.001	<0.001
	5 mm	3.75	3.75	4.13	4.39	4.64	0.02				<0.001	0.054
Kibble bulk density after extruder (g/L)												
	7 mm	471	507	537	530	527	1.63	<0.001	<0.001	<0.001	<0.001	<0.001
	5 mm	332	346	324	342	342	1.69				0.036	0.189
Piece density (g/cm^3^)												
	7 mm	0.43	0.64	0.84	0.82	0.78	0.006	<0.001	<0.001	<0.001	<0.001	<0.001
	5 mm	0.40	0.43	0.43	0.44	0.46	0.006				0.058	0.662
Cutting force (N)												
	7 mm	21.9	29	72.2	74.6	59.7	0.81	<0.001	<0.001	<0.001	<0.001	<0.001
	5 mm	21.3	26.9	26.4	30.1	29.2	0.81				<0.001	<0.001
Starch gelatinization degree (%) ^e^												
	7 mm	90	89.9	80.8	81.4	82.5	0.71	<0.001	<0.001	<0.001	<0.001	0.003
	5 mm	95.5	94.8	92.9	94	89.6	0.82				<0.001	<0.001
In vitro digestibility of organic matter												
	7 mm	0.82	0.79	0.74	0.75	0.73	0.005	0.001	<0.001	<0.001	<0.001	0.013
	5 mm	0.81	0.80	0.78	0.76	0.73	0.005				<0.001	0.156

^a^ CO = control food, without added citrus pulp pellet; CPP5 = addition of 5% citrus pulp pellet; CPP10 = addition of 10% citrus pulp pellet; CPP15 = addition of 15% citrus pulp pellet; CPP20 = addition of 20% citrus pulp pellet. ^b^ SEM = standard error of the mean (n = 20). ^c^ d_d_ = diet diameter; CPP = citrus pulp pellet. ^d^ Linear and quadratic effect of citrus pulp pellet addition. ^e^ die diameters = d_d_ of 5 mm (resulting in a die open area of 19.6 mm^2^) and d_d_ of 7 mm (resulting in a die open area of 38.5 mm^2^).

## Data Availability

Not applicable.

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
