# Peer review of "Effect of Citrus Pellet on Extrusion Parameters, Kibble Macrostructure, Starch Cooking and In Vitro Digestibility of Dog Foods"

_animals, 2023, doi:10.3390/ani13040745_

Round 1
Reviewer 1 Report
I enjoyed reading this paper and I think it brings great contribution to the scarce literature on pet food processing. The work needs minor revisions, and I found a few grammatical issues that have been addressed below. I also made a few suggestions to improve data presentation and a few considerations to add to the discussion. Great job overall!
L17- 18 “The aim of the study was to evaluate the influence of increasing amounts of CPP on extrusion traits,..”
You simple summary is very similar to the abstract, with some parts identical. It can be simplified more, and it needs to be less technical- think of it as an explanation to lay people.
L29- The aim..
L50-51: “which are rich in fermentable carbohydrates mostly composted of soluble dietary fiber, such as pectin.”
L51- “fiber source”
L53: “for promoting a
L54: “in the distal part of the gastrointestinal tract,”
L57: “and differentiation, and..”
L61: “..able fiber that promote a good fermentation..”
L69: “dough expands upon exiting the barrel at atmospheric pressure, creating..”
L73-75: Please add one or more references
L78: You may also substitute “starch cooking” with “starch cook” throughout the paper, as the latter is the most used term
L92 & L104- Brazil
L94-95 “.., without added fiber source, was…”
Table 2: Poultry by-product meal
Table 2: suggestion to add two subheadings: “dry mix” or “dry formula”, and “coating”
L114- hammermill
L125: No water or steam was injected in the extruder barrel
L127- double-shaft..
L129: ...an average mass retention time of 182.7s per diet at the preconditioner.
L135: …at the preconditioner.. – also, remove (kg/h) because you already specified the unit
L140: ..at two different days..
L140 suggestion: The CO diet was fed to the extruder at the beginning of both days to warm up the equipment and reach steady state before starting treatment production.
L154: until further analyses.
L146: I don’t think you can state that there were 8 observations per treatment, because it was a factorial design (5 diets x 2 die diameters) with 4 observations in each combination
L155: liquid palatant
L159: ..were: CO, 285 and 298μm; CPP5, 278 and 263μm; …
L160: is the mean geometric diameter of of 287μm the overall mean of all treatments combined? If so, write “.., with an overall mean geometric diameter..”
L162- Chemical analyses
L181- at pH 2
L210- substitute extrudates with kibbles
L212- should it be de2/dd2?
L216: extruder or extrudate ?
L219- .. using a texture analyzer (TA(T?)X/T21…
L236- which one did you consider the experimental unit? The day of extrusion or the sample and processing data collected at each 15-min interval subplots? It seems that you considered the subsamples collected at 15 min the replicates? Not ideal, but understandable for this type of processing work since we usually only have one extruder. Since you used one die diameter in one day and another die diameter the other day, I am not sure the day would be the EU. I think the extruder would be the EU and the subsamples (collected every 15 min), the replicates. Please consult with your statistician.
L241- would it be “orthogonal polynomial contrast”?
242- how did you check that the data complied with the ANOVA model, did you analyze studentized residuals?
L 245- remove “sample used”
L246- which were expected given the ingredient characteristics (Table 1)
L 247-248 …was the substitution of maize by increasing additions of CPP, without altering the other raw materials’ inclusion”
251- among treatments.
Table 3- where is TDF?
Table 3- looking at CP, we could see a trend of protein decreasing as CPP was added, but at CPP15 the protein level when produced with the 5mm die diameter was pretty high, any explanation for that, or do you think it was analytical variation?
L258- you mention that IBM did not change enough to affect the products extruded using the 5 and the 7mm diameter. However, when produced with the 5mm, the IBM was significantly higher with most CPP inclusions than the 7mm IBM. Do you have an explanation of why this happened?
The order of parameters you mention under Results does not follow the order that appear on the Table. It would be preferred that they followed the same order, so maybe rearranging the Table would help.
L260- I don’t understand what you mean by “results within a treatment condition”- would it read better if it was “.. influence treatment results”?
L261- I don’t understand what you mean by “within one extrusion condition”
L263- Substitute “for the processing condition of 7 mm dd” for “while producing diets with the 7 mm dd”
L264-267- “As proposed on the experimental design, the differences in dd caused the extruder open area to be more than 50% greater when the 7 mm dd was used in comparison to the 5 mm dd (p < 0.01).” – This part was confusing, please see if you agree with the change.
L267-269- The higher restriction to the extruder mass flow explains the greater application of specific mechanical energy observed for the 5 mm Dd compared to the 7 mm dd extruder configuration (p < 0.01).
Paragraph 277-281- you need to cite Table 4 again here at some point. You don’t need to cite the linear changes if there were quadratic changes.
L282- remove “of” and substitute “formation” with “characteristics”
L286- “..of mechanical energy with the 5 mm...”
L285-290- this phrase is too long and difficult to read- it needs to be divided in a few sentences.
L295-297- “Another example is the cutting force, or kibble hardness, that increased quadratically in both dd (p < 0.01), but kibbles produced with the 7 mm dd configuration were harder (p < 0.01).”
Table 5- same as Table 4, the written part of Results did not follow the order that parameters appear in the Table. Also, Table 5 needs to be cited in the paragraph 307-313.
L318- verify that the number of cells per kibble increased (with the addition of CPP?).
L342- has been previously..
L343- and in a recent publication after the CPP inclusions in dogs [4]. – do you mean dogs’ snacks? If so, please simplify: “in human (refs) and dogs’ extruded snacks (ref).
Discussion- first paragraph- please add a reference for the fibers composition. Very interesting observation on the different types of fiber on extrusion parameters.
L368- “when there is sufficient water, energy, and time during processing,..” Starch granules lose their crystallinity,… Still on starch, it’s important to mention the relevance of using the 5mm vs the 7mm die, as the 5mm restricted mass flow and caused an increase in energy inside the barrel, which allowed starch to gelatinize.
L370- that binds the food components in a continuous structure.
L372-373- interesting point. Worth recalling that the main fiber component is soluble (pectin?), and that’s why it retains water.
It would be good to mention which extruder parameters you aimed to keep constant across treatments. If it had been bulk density, you would have made changes to target a similar expansion. Could you please add a table with extruder fixed effects that you used? Such as screw speed, knife speed, feed rate, etc.
L382- cell size
L392- that was sufficient in increasing..
L393- Furthermore, kibble bulk density increased after the CPP addition when using the 7 mm dd, but..
L396- ..altering extrusion processing conditions such as die diameter.
L408- what do you mean by “internal cells dropped”?
L409- “after increasing die restriction and subsequently SME”
L412- on the surface. The fat distribution (inside?) the kibbles
L426- increasing the use of fiber-supplemented diets..
Paragraph 426-432- I would like to see some mention of the benefits to the gut microbiome with the addition of CPP. Both due to the presence of soluble fiber in CPP, as well as due to the alteration in starch cook that this fiber caused, which would increase the amount of resistant starch present in the food. These are important aspects for the industry, as we aim to increase our pets’ longevity. Also, did you conduct a palatability test? This would be a very interesting addition.
L439-440- Recycling and properly utilizing co-products in pet foods, such as CPP exemplified in the present study, also improves agricultural profitability..
L441- you need to pick one term- either co-products or by-products
L448- what do you mean by “limited processing efficiency”? did you have a greater material loss due to surging because of the fiber? I thought the CPP decreased energy expenditure of the extruder.
L467- for the donation of the extruder
Author Response
Reviewer 1
I enjoyed reading this paper and I think it brings great contribution to the scarce literature on pet food processing. The work needs minor revisions, and I found a few grammatical issues that have been addressed below. I also made a few suggestions to improve data presentation and a few considerations to add to the discussion. Great job overall!
Dear Reviewer. Thank you very much for your positive evaluation of our contribution! We also acknowledge your dedication to reviewing and improving the text. We considered all your suggestions and answered to them below.
L17- 18 “The aim of the study was to evaluate the influence of increasing amounts of CPP on extrusion traits,..” done
You simple summary is very similar to the abstract, with some parts identical. It can be simplified more, and it needs to be less technical- think of it as an explanation to lay people. Done (line 13-25)
L29- The aim.. done (line 28)
L50-51: “which are rich in fermentable carbohydrates mostly composted of soluble dietary fiber, such as pectin.” Done (line 49)
L51- “fiber source” done (line 50)
L53: “for promoting a done (line 52)
L54: “in the distal part of the gastrointestinal tract,” done (line 53)
L57: “and differentiation, and..” done (line 57)
L61: “..able fiber that promote a good fermentation..” done (line 60)
L69: “dough expands upon exiting the barrel at atmospheric pressure, creating..” done (line 69)
L73-75: Please add one or more references done (line 78; Souza et al., 2022)
L78: You may also substitute “starch cooking” with “starch cook” throughout the paper, as the latter is the most used term done (line 81, 374, 403, 446, 449, 467, 472)
L92 & L104- Brazil done (lines 104 and 108)
L94-95 “.., without added fiber source, was…” done (line 94)
Table 2: Poultry by-product meal done
Table 2: suggestion to add two subheadings: “dry mix” or “dry formula”, and “coating” done
L114- hammermill done (line 119)
L125: No water or steam was injected in the extruder barrel done (lines 130, 131)
L127- double-shaft.. done (line 132)
L129: ...an average mass retention time of 182.7s per diet at the preconditioner. Done (line 134)
L135: …at the preconditioner.. – also, remove (kg/h) because you already specified the unit done (line 136)
L140: ..at two different days.. done (line 144)
L140 suggestion: The CO diet was fed to the extruder at the beginning of both days to warm up the equipment and reach steady state before starting treatment production. Done (lines 146-147)
L154: until further analyses. Done (lines 159-160)
L146: I don’t think you can state that there were 8 observations per treatment, because it was a factorial design (5 diets x 2 die diameters) with 4 observations in each combination
Dear Reviewer. In the factorial design we had 5 formulations and 2 processing condition. The diets were produced in 2 separate days, and in each day 4 replications (time of collection) were applied. This procedure resulted in 8 replications in total, 4 on the first and 4 on the second day of extrusion. Thank you. (lines 154-156)
L155: liquid palatant done (line 164)
L159: ..were: CO, 285 and 298μm; CPP5, 278 and 263μm; … done (lines 169-170)
L160: is the mean geometric diameter of of 287μm the overall mean of all treatments combined? If so, write “.., with an overall mean geometric diameter..” done (line 170)
L162- Chemical analyses done (line 172
L181- at pH 2 done (line 189)
L210- substitute extrudates with kibbles done (line 219)
L212- should it be de2/dd2? Done (line 222)
L216: extruder or extrudate ? done (lines 227-228)
L219- .. using a texture analyzer (TA(T?)X/T21… done (line 228)
L236- which one did you consider the experimental unit? The day of extrusion or the sample and processing data collected at each 15-min interval subplots? It seems that you considered the subsamples collected at 15 min the replicates? Not ideal, but understandable for this type of processing work since we usually only have one extruder. Since you used one die diameter in one day and another die diameter the other day, I am not sure the day would be the EU. I think the extruder would be the EU and the subsamples (collected every 15 min), the replicates. Please consult with your statistician.
Dear Reviewer, perhaps we have not explained ourselves well in section 2.2 (Diet preparation). We extruded each food twice. So, we have now included: “All diets (CO, CPP5, CPP10, CPP15, CPP20) were processed separately on two different days for each dd, for a total of two extrusion days for each combination of CPP and dd level” . Then the extrusion day (2 per diet*dd combination) was the plot and the sampling time at each 15 minute interval (4 per run) the subplot in the design. With this clarification in section 2.2 we hope it is now clear. (lines 118-124)
L241- would it be “orthogonal polynomial contrast”? done (line 249)
242- how did you check that the data complied with the ANOVA model, did you analyze studentized residuals?
We included: error normality was evaluated by the Cramer-von Mises test and homoscedasticity of variance by the Levene test) (lines 250-252)
L 245- remove “sample used” done
L246- which were expected given the ingredient characteristics (Table 1) done (line 256)
L 247-248 …was the substitution of maize by increasing additions of CPP, without altering the other raw materials’ inclusion” done (lines 257-258)
251- among treatments. Done (Line 259)
Table 3- where is TDF?
Dear reviewer, unfortunately we did not evaluate TDF for the experimental diets, only CPP (Table 1).
Table 3- looking at CP, we could see a trend of protein decreasing as CPP was added, but at CPP15 the protein level when produced with the 5mm die diameter was pretty high, any explanation for that, or do you think it was analytical variation?
We have noticed this too. The diets were produced with the same batch of raw materials, but we sometimes have variations in the CP content of the poultry by-product meal or in the efficiency of sampling, mixing and nitrogen analysis.
L258- you mention that IBM did not change enough to affect the products extruded using the 5 and the 7mm diameter. However, when produced with the 5mm, the IBM was significantly higher with most CPP inclusions than the 7mm IBM. Do you have an explanation of why this happened?
Every effort has been made to have as little variation in in-barrel moisture between treatments as possible. These variations were the result of differences in internal moisture of raw material and fluctuations in "extruder inputs", such as variations in dry matter feed rate, water flow to the preconditioner, and steam flow and absorption in the mass (steam can “scape” from preconditioner without condensing into the mass). Our comment was that while there was a change noted, there was little rallying to change the processing substantially.
The order of parameters you mention under Results does not follow the order that appear on the Table. It would be preferred that they followed the same order, so maybe rearranging the Table would help. done
L260- I don’t understand what you mean by “results within a treatment condition”- would it read better if it was “.. influence treatment results”? done (lines 264-265)
L261- I don’t understand what you mean by “within one extrusion condition” done
L263- Substitute “for the processing condition of 7 mm dd” for “while producing diets with the 7 mm dd” Explained better (llines 265-268)
L264-267- “As proposed on the experimental design, the differences in dd caused the extruder open area to be more than 50% greater when the 7 mm dd was used in comparison to the 5 mm dd (p < 0.01).” – This part was confusing, please see if you agree with the change. Done (lines 268-273)
L267-269- The higher restriction to the extruder mass flow explains the greater application of specific mechanical energy observed for the 5 mm Dd compared to the 7 mm dd extruder configuration (p < 0.01). done (lines 271-273)
Paragraph 277-281- you need to cite Table 4 again here at some point. You don’t need to cite the linear changes if there were quadratic changes. Done (line 275 and 278)
L282- remove “of” and substitute “formation” with “characteristics” done (line 279)
L286- “..of mechanical energy with the 5 mm...” done (line 284)
L285-290- this phrase is too long and difficult to read- it needs to be divided in a few sentences. Done (lines 284-287)
L295-297- “Another example is the cutting force, or kibble hardness, that increased quadratically in both dd (p < 0.01), but kibbles produced with the 7 mm dd configuration were harder (p < 0.01).” done (lines 293-295)
Table 5- same as Table 4, the written part of Results did not follow the order that parameters appear in the Table. Also, Table 5 needs to be cited in the paragraph 307-313. done
L318- verify that the number of cells per kibble increased (with the addition of CPP?). done
L342- has been previously. Done (line 351)
L343- and in a recent publication after the CPP inclusions in dogs [4]. – do you mean dogs’ snacks? If so, please simplify: “in human (refs) and dogs’ extruded snacks (ref). done (lines 351-352)
Discussion- first paragraph- please add a reference for the fibers composition. Very interesting observation on the different types of fiber on extrusion parameters. Done (line 354; Nieto et al., 2021)
L368- “when there is sufficient water, energy, and time during processing,..” Starch granules lose their crystallinity,… done (line 377)
Still on starch, it’s important to mention the relevance of using the 5mm vs the 7mm die, as the 5mm restricted mass flow and caused an increase in energy inside the barrel, which allowed starch to gelatinize.
We agree, thanks for this comment. In fact, the modulation of SEM transference through the reduction of the die diameter was the primary intention of our observation. We included: “In the present study, when a higher restriction was applied in the extruder die (5 mm dd), the increase in mass flow resistance increased energy application, restoring starch transformation and cooking” ines 379-381
L370- that binds the food components in a continuous structure. Done (line 379)
L372-373- interesting point. Worth recalling that the main fiber component is soluble (pectin?), and that’s why it retains water. Done (line 384)
It would be good to mention which extruder parameters you aimed to keep constant across treatments. If it had been bulk density, you would have made changes to target a similar expansion. Could you please add a table with extruder fixed effects that you used? Such as screw speed, knife speed, feed rate, etc.
Dear Reviewer, The intention of this sentence is only to comment to the reader that many aspects of extrusion parameters should be explored to overcome the difficulties of incorporating fiber into kibble diets. In the present study we explored one, resistance to mass flow (reducing the open area), but others should be tried in other studies as well (some are listed in parentheses). We think you don't need a table or longer text for this message. The parameters we have set are all described and explained in the materials and methods section. Our goal was not similar bulk densities, as this would require many changes in several parameters compromising the objectivity of scientific observation. Our intention was to describe the impact of adding CPP on bulk density. Thank you.
L382- cell size done (line 392)
L392- that was sufficient in increasing.. done (line 403)
L393- Furthermore, kibble bulk density increased after the CPP addition when using the 7 mm dd, but.. done (line 405)
L396- ..altering extrusion processing conditions such as die diameter. Done (line 407)
L408- what do you mean by “internal cells dropped”?
It wasn't clear. We rephrased, Thank you. (line 418)
L409- “after increasing die restriction and subsequently SME” done (lines 419-420)
L412- on the surface. The fat distribution (inside?) the kibbles done (lines 420-422)
L426- increasing the use of fiber-supplemented diets.. done (line 425)
Paragraph 426-432- I would like to see some mention of the benefits to the gut microbiome with the addition of CPP. Both due to the presence of soluble fiber in CPP, as well as due to the alteration in starch cook that this fiber caused, which would increase the amount of resistant starch present in the food. These are important aspects for the industry, as we aim to increase our pets’ longevity. Done (lines 442-445)
Also, did you conduct a palatability test? This would be a very interesting addition.
Would be very nice. But unfortunately, we haven't done that in this process.
L439-440- Recycling and properly utilizing co-products in pet foods, such as CPP exemplified in the present study, also improves agricultural profitability.. done (lines 458-459)
L441- you need to pick one term- either co-products or by-products done (line 460)
L448- what do you mean by “limited processing efficiency”? did you have a greater material loss due to surging because of the fiber? I thought the CPP decreased energy expenditure of the extruder.
We meant that the limitation in the SME transference also compromises cooking, with less gelatinization of the starch. We rephrased to improve the message. (Line 467)
L467- for the donation of the extruder done (line 488)

Reviewer 2 Report
It is a good paper in terms of scientific rightness, very technical and well written, having, in my view, limited interest for not including data regarding in vivo digestabilitily or palatatibility.
Author Response
Reviewer #2
It is a good paper in terms of scientific rightness, very technical and well written, having, in my view, limited interest for not including data regarding in vivo digestabilitily or palatatibility.
Dear Reviewer, Thank you for your positive evaluation of our contribution. In our view, scientific interest varies between different research perspectives. For those interested in animal (dog) nutrition and ingredient usage, we agree that interest may be limited by the lack of animal data. However, other readers come from areas of processing technology and to them this detailed and in-depth exploration of the rheological aspects of incorporating an ingredient into a kibble diet is of interest. Thank you very much.

Reviewer 3 Report
I'd like to congratulate the Authors on fine work. My few questions and comments would be:
'energy transference' - in your paper you use this term quite often. I tried to check in Pacheco et al. 2018 and 2021 for a clear explanation and found a single occurence (in abstract) in each. My suggestion would be to add a clear explanation/definition early in the text.
Line 38: 'agriculture profitability' - can you generalize to this extent? or would it be more particular, smaller area?
Line 63: 'During extrusion to produce dog food...' - do you mean 'during the extrusion phase in dry dog food production process'?
Line 268: an abbreviation (SME) should be used;
Line 292: (non statistical) - did you mean 'not significant'?
Author Response
Reviewer 3
Dear Reviewer, Thank you very much for your time, evaluating our contribution. We are happy about your positive feedback. All your suggestions have been taken into consideration and answered below.
I'd like to congratulate the Authors on fine work. My few questions and comments would be:
'energy transference' - in your paper you use this term quite often. I tried to check in Pacheco et al. 2018 and 2021 for a clear explanation and found a single occurence (in abstract) in each. My suggestion would be to add a clear explanation/definition early in the text. Done (lines 74-76)
Line 38: 'agriculture profitability' - can you generalize to this extent? or would it be more particular, smaller area? Done (line 37)
Line 63: 'During extrusion to produce dog food...' - do you mean 'during the extrusion phase in dry dog food production process'? The sentence was changed (lines 63-70)
Line 268: an abbreviation (SME) should be used; done (line 278)
Line 292: (non statistical) - did you mean 'not significant'? done (line 290)
